# ChIP-Seq-Based Approach in Mouse Enteric Precursor Cells Reveals New Potential Genes with a Role in Enteric Nervous System Development and Hirschsprung Disease

**DOI:** 10.3390/ijms21239061

**Published:** 2020-11-28

**Authors:** Leticia Villalba-Benito, Ana Torroglosa, Berta Luzón-Toro, Raquel María Fernández, María José Moya-Jiménez, Guillermo Antiñolo, Salud Borrego

**Affiliations:** 1Department of Maternofetal Medicine, Genetics and Reproduction, Institute of Biomedicine of Seville (IBIS), University Hospital Virgen del Rocío/CSIC/University of Seville, 41013 Seville, Spain; leticia.villalba.benito@hotmail.es (L.V.-B.); ana.torroglosa@juntadeandalucia.es (A.T.); berta.luzon@ciberer.es (B.L.-T.); raquelm.fernandez.sspa@juntadeandalucia.es (R.M.F.); gantinolo@us.es (G.A.); 2Centre for Biomedical Network Research on Rare Diseases (CIBERER), 41013 Seville, Spain; 3Department of Pediatric Surgery, University Hospital Virgen del Rocío, 41013 Seville, Spain; mariajosemoyajimenez@gmail.com

**Keywords:** Hirschsprung disease, ChIP-seq, gene expression profiling, sequence analysis, *PAX6*

## Abstract

Hirschsprung disease (HSCR) is a neurocristopathy characterized by intestinal aganglionosis which is attributed to a failure in neural crest cell (NCC) development during the embryonic stage. The colonization of the intestine by NCCs is a process finely controlled by a wide and complex gene regulatory system. Several genes have been associated with HSCR, but many aspects still remain poorly understood. The present study is focused on deciphering the PAX6 interaction network during enteric nervous system (ENS) formation. A combined experimental and computational approach was performed to identify PAX6 direct targets, as well as gene networks shared among such targets as potential susceptibility factors for HSCR. As a result, genes related to PAX6 either directly (*RABGGTB* and *BRD3*) or indirectly (*TGFB1*, *HRAS*, and *GRB2*) were identified as putative genes associated with HSCR. Interestingly, *GRB2* is involved in the RET/GDNF/GFRA1 signaling pathway, one of the main pathways implicated in the disease. Our findings represent a new contribution to advance in the knowledge of the genetic basis of HSCR. The investigation of the role of these genes could help to elucidate their implication in HSCR onset.

## 1. Introduction

Hirschsprung disease (HSCR, OMIM 142623) or aganglionic megacolon, is a disorder characterized by intestinal aganglionosis in a variable segment of the distal gut, whose main manifestation is constipation or intestinal obstruction [1]. The cause of HSCR is attributed to a defective proliferation, differentiation, survival, and/or migration of enteric precursor cells (EPCs) originating from the neural crest between the seventh and twelfth weeks of gestation. HSCR can be classified into three types based on the length of the aganglionic tract, including short-segment HSCR (S-HSCR), long-segment HSCR (L-HSCR), and total colonic/intestinal aganglionosis (TCA/TIA), which represents around 80%, 15%, and 5% of the cases, respectively. In addition, HSCR can appear either sporadically or within a familial aggregation and may be associated with other developmental defects [2]. Alterations in the specific gene expression patterns required at different stages of the enteric nervous system (ENS) formation can lead to HSCR [3]. To date, a large number of different genes have been related to HSCR. In most cases, these genes encode molecules that are widely involved in the major signaling pathways associated with HSCR, the most important being the RET/GDNF/GFRA1 and EDN3/EDNRB signaling pathways [4]. Nevertheless, the genetic cause of the disease in a large percentage of HSCR patients remains unknown; thus, the search for additional genes and mechanisms to better outline HSCR etiology is still required. HSCR, especially the short segment forms, is a disorder with a complex genetic basis that could be explained by the heterogeneity of ENS development, which is regulated by an ever-increasing range of molecules. In accordance with this, the HSCR phenotype is often the result of the presence of various genetic variants, which contribute in an additive manner [5]. Despite the fact that some families show an either dominant or recessive pattern, most of the cases show a complex and non-Mendelian mode of inheritance with variable penetrance [2,5].

One of the best-known mechanisms that controls gene expression during ENS development is the regulation carried out by transcription factors. In this sense, several studies have identified transcription factors expressed during the different stages of the ENS development [6,7]. A few transcription factors have been previously associated with HSCR onset, such as ZFXH1B, SOX10, PHOX2B, and PAX6 [8,9,10,11,12], which modulate expression of different downstream effectors at the developmental stage where EPCs migrate along the gut. Specifically, the expression of *PAX6* both at the mRNA and the protein levels in EPCs was previously detected by our group [12,13]. Reduced *PAX6* expression levels were detected in HSCR patients and different mechanisms that might lead to this downregulation have been proposed. This study suggested that *PAX6* may have a role in the regulation of the transcriptional network involved in ENS development [12]. Therefore, the aims of the present study were to clarify the specific gene expression pattern established by PAX6 for correct ENS development, as well as to identify new susceptibility genes for HSCR. With this purpose, we performed a chromatin immunoprecipitation coupled with a massively parallel sequencing (ChIP-seq) assay to identify PAX6 target genes in mouse EPCs, as well as a differential expression assay in EPCs from HSCR patients versus controls to evaluate their implication in the onset of the disease. In addition, we have searched for rare sequences that may be associated with this pathology within the coding sequence of the resulting genes that we found to be expressed in human EPCs.

## 2. Results

### 2.1. Identification of the PAX6 Target Genes in Mouse Neurosphere-Like Bodies (NLBs)

EPCs grow in culture as cell aggregates known as neurosphere-like bodies (NLBs), which are a population of neural crest-derived multipotent stem cells present in postnatal gut. The current study was focused on the identification of PAX6 gene regulatory networks during ENS development to detect additional genes and mechanisms to better outline HSCR etiology. With this aim, a ChIP-seq assay was performed using primary culture NLBs from mouse. The reason for this is that a considerable starting quantity of DNA is needed for this technique and that only one sample for each patient was available. 

Upon completion of sequencing, the percentage of reads that passed the internal quality filtering, defined as “pass filter reads” (PF Reads) was >90% (27.2 × 10^6^ reads). A total of 16.8 x 10^6^ reads mapped with the reference genome. As result, a total of 2247 peaks were detected as well as a total of 4250 genes as the nearest genes to the peaks. Therefore, a methodological approach based on the one described in Villalba–Benito et al. [14] was used for the purpose of selecting a handful of putative targets as well as the most reliable genes suitable to be HSCR candidate genes (pipeline described in the Methods Section). Application of the data analysis pipeline for ChIP-Seq allowed for the identification of 19 PAX6 target genes that resulted from 17 different PAX6 binding regions (Table 1). To further characterize these 17 PAX6 binding regions, we investigated the presence of possible novel PAX6 DNA-binding motifs into these sequences using the MEME-ChIP tool (motif discovery). This analysis revealed two possible biologically relevant motifs that were over-represented (Table 2). 

### 2.2. Selection of Genes to Evaluate Their Implication in ENS Development and HSCR

To determine the involvement of PAX6 target genes in human ENS development, we searched for the human orthologs genes of the PAX6 target genes identified in mice by the Mouse Genome Informatics database (MGI). It should be noted that the human orthologs gene for *Speer5-ps1*, *Gm15997*, and *Wdr95* still remain unknown. Therefore, we eliminated these genes from the analysis. Afterwards, gene interactions of the PAX6 target genes by the Ingenuity Pathway Analysis (IPA) tool were analyzed. Our main aim was to identify new potential susceptibility genes for HSCR and thus, after analyzing them, we decided to select 10 additional genes (*TGFB1, VHL, APP, MYC, NTRK1, GRB2, HRAS, HTT, TNF*, and *HIST1H3A*) based on their interaction with several PAX6 target genes (Appendix A). Therefore, a total of 26 genes were considered to further assess their potential implication in ENS development and HSCR (16 PAX6 target genes and 10 related genes) (Table 3).

### 2.3. Gene Expression Patterns in NLBs From HSCR Patients Versus NLBs From Controls

To identify if the selected genes were involved in human ENS development, their gene expression profiles were analyzed using NLBs from the human gut. We observed expression of 11 genes, 5 of them being PAX6 target genes (*ACADM*, *ATXN1*, *BRD3*, *COL4A2*, *RABGGTB*) and the remaining genes being related to them (*APP*, *GRB2*, *HRAS*, *HTT, MYC*, *TGFB1*) (Appendix A). It is worth mentioning the identification of *Edn3* as a target gene of PAX6 since EDN3 is a member of one of the main pathways implicated in HSCR. We did not detect expression in human NLBs, which agrees with the expression pattern of *EDN3* in the surrounding mesenchyme of ENS. However, our interest in this study was not the study of well-known genes related to the disease, rather than detecting new genes involved in such pathology. Thus, we focused on following up the study of new candidate genes. 

To assess the possible role of these genes in the pathogenesis of HSCR, their expression levels in NLBs from HSCR patients versus controls were compared. We identified 5 genes with statistically significant different expression levels (*GRB2*, *BRD3*, *HRAS*, *RABGGTB*, and *TGFB1*; *p*-value ≤ 0.05). Specifically, *HRAS*, *RABGGTB*, and *TGFB1* were upregulated, whereas *GRB2* and *BRD3* were downregulated in HSCR-NLBs (Figure 1).

### 2.4. Identification and Assessment of Susceptibility Rare Variants for HSCR in the Genes Expressed in Human ENS

The sequences of the genes that showed expression in human NLBs (*ACADM*, *ATXN1*, *BRD3*, *COL4A2*, *RABGGTB*, *APP*, *GRB2*, *HRAS*, *HTT*, *MYC*, *TGFB1*) were analyzed to identify potential pathogenic variants related to HSCR, through the analysis of the whole exome sequencing (WES) data from 56 HSCR patients [15]. As a result, a set of rare variants were identified in *ATXN1*, *RABGGTB*, *COL4A2*, *HTT*, *APP*, and *GRB2*. All these rare variants were inherited from one of their parents and occasionally they appeared in an unaffected sibling; therefore, it seems that these rare variants do not play an important role in HSCR susceptibility. Such variants are available under request. 

### 2.5. Functional Networks Shared among the Possible Susceptibility Genes for HSCR

To determine the functional association networks among the genes identified as putative susceptibility genes in HSCR, we used GeneMANIA software (Appendix A). As a result, all the genes interacted with each other through physical interactions, co-expression, and pathway. In addition, gene function prediction showed that *TGFB1*, *HRAS*, and *GRB2* contribute to the ErbB or epidermal growth factor receptor (EGFR) signaling pathway. *HRAS* and *TGFB1* also showed functions related to cell cycle and cell migration regulation.

### 2.6. GRB2 Implication in the Onset of HSCR

Based on our results and the fact that *GRB2* encodes for the intracellular adaptor protein growth factor receptor-bound 2, which is involved in RET/GDNF/GFRA1 signaling, we decided to further investigate the association of *GRB2* with HSCR onset. We performed a mutational screening by direct sequencing of the *GRB2* gene, including its coding sequence, intron/exon boundaries and untranslated regions (UTR) in 267 isolated HSCR patients. As a result, we identified several *GRB2* rare variants that alone do not seem to significantly contribute to the manifestation of the phenotype, ruling out, therefore, the presence of deleterious mutations in this gene as a mechanism leading to HSCR. The variants identified in GRB2 are available on request. 

To investigate the function of GRB2 in the onset of HSCR, *Grb2* expression was downregulated in NLBs from mouse. *Grb2* expression was reduced by 30% compared to the control. It is important to mention that this reduction was very similar to *GRB2* downregulation found in NLBs from HSCR patients. First, we analyzed the effect of *Grb2*-knockdown (LV-*Grb2*) on NLBs’ growth, but no changes in size or number of NLBs were identified. Then, we performed an expression analysis of markers associated with the enteric precursor (p75), neural precursor (Nestin) and neuronal precursor cell (βlll-Tubulin). As a result, LV-*Grb2* cultures showed a soft decline of Nestin^+^ cells compared to both the uninfected group (control) and non-target control group (LV-off-target) conditions (12% and 14.9% respectively). Data are shown in Figure 2.

## 3. Discussion

Neural crest-derived cells give rise to ENS through a complex process carefully regulated by interacting signals and a specific gene expression pattern. It is accepted that failures in these processes are responsible for the incomplete gut colonization by these cells during embryonic development, resulting in HSCR. A deeper knowledge about transcriptional programs required for EPCs development, might help us to identify new genes, pathways, and mechanisms involved in HSCR pathogenesis. Specifically, *PAX6* was described by our group as a potential susceptibility gene for HSCR, suggesting that it could have a regulatory role on the transcriptional network during the ENS development [12]. Thus, we have focused on the identification of the gene expression pattern established by PAX6 and their potential implication in the ENS development and in HSCR.

In the current study, we identified 11 genes with a potential role during human ENS development based on their expression in human NLBs. *ACADM*, *COL4A2*, *RABGGTB*, *ATXN1*, and *BRD3* were identified as PAX6 targets and *HRAS*, *HTT*, *MYC*, *APP*, *TGFB1*, and *GRB2* as genes related to such targets. In addition, the approach carried out revealed a group of genes that might be involved in the onset of HSCR, being good candidates to further investigate their role during ENS development (*GRB2*, *BRD3*, *HRAS*, *RABGGTB*, and *TGFB1*). The workflow with the selection, prioritization, and further validation of the candidate genes is shown in Figure 3. 

Transcription factors bind to short 5–20 bp segments of DNA to regulate gene expression. Such regulatory sequences can be discovered computationally by a computational algorithm that looks for a short, conserved, and often repeated pattern called “the motif”. Such motifs are likely to be the transcription factor binding sites [16]. We detected two potential biologically relevant sequences over-represented in the PAX6 binding sequences. 

Our results about *Edn3* as a PAX6 target allowed us to validate the experimental approach performed in the present study to identify new susceptibility genes for HSCR, as well as to reinforce the role of PAX6 during ENS development. This study has prompted us to examine in detail in future studies the regulatory mechanism of PAX6 over *EDN3* since, given that transcription factors can inhibit gene expression [17,18,19], PAX6 might be playing an inhibitory role on *EDN3*.

The detection of *GRB2* as a new candidate gene for HSCR should be mentioned. *GRB2* encodes for the intracellular adaptor protein growth factor receptor-bound 2, which is involved in RET/GDNF/GFRA1 signaling. *RET* activation results in phosphorylation of key docking tyrosines that bind to several intracellular adaptor proteins such as GRB2, and therefore, this protein acts as a regulator of this signaling pathway during ENS development [20]. GRB2 can directly bind to RET Y1096 and indirectly to RET Y1062, leading to PI3K/AKT or MAPK signaling activation [21]. A variant at Y1062 caused distal colon aganglionosis reminiscent of HSCR in mice and it has also been described in HSCR patients [22,23]. Based on our data, *Grb2* knockdown in ENS precursors resulted in a slight reduction in Nestin-expressing cells (neural precursor cell), suggesting that these enteric precursors could be in a more differentiated state as compared with the control. However, we did not observe any difference in the number of beta-tubulin III-expressing cells, suggesting that *Grb2* has no effect on neuronal differentiation. Some studies have described a role of *Grb2* that affects the cellular processes related to glial cells [24,25]. Therefore, although further studies are needed, it seems that *Grb2* might have a role in glial differentiation. 

*RABGGTB* showed differential expression in NLBs from HSCR patients. *RABGGTB* is highly expressed between E11.5 and E13.5 during mouse embryos’ development [26,27]. In mice, neural crest cells (NCCs) reach the region of the caecum at E11.5 and completely colonize the intestine at E14.5 [28].

It is also interesting to notice the identification of *APP* as a gene with a potential role during human ENS development. According to our study, this gene was previously identified to be highly expressed in enteric NCCs together with *Ret* and *Sox10*, as well as a key regulator of ENS development [29], and a potential gene associated with HSCR [30].

In our study, we have observed that these potential susceptibility genes for HSCR were functionally connected to each other. In particular, *TGFB1*, *HRAS*, and *GRB2* were predicted to be involved in the ERBB signaling pathway, which plays an important role in the development of ENS [31]. Moreover, some genes implicated in this pathway (*NRG1* and *NRG3*) have already been identified as susceptible genes for HSCR [32,33,34]. These functional connections further support the role of these candidate genes in ENS formation.

In conclusion, we deepened the understanding of the expression pattern regulated by PAX6 during the ENS development and determined a set of genes directly regulated by PAX6 through specific regulatory binding sequences. Moreover, we showed a group of genes with a potential implication in ENS development and in HSCR onset. Among them, we identified *GRB2* and its direct association with HSCR for the first time. The application of the approach carried out in this study can be an effective method for the knowledge of the genetic background of other rare disorders with a complex genetic basis. 

## 4. Materials and methods 

### 4.1. Generation of Enteric NLBs

The isolation of EPCs from postnatal gut represents a powerful tool for the study of ENS development. In culture, these cells form aggregates in suspension known as NLBs, which contain neural crest-derived stem cells and their progeny. Enteric NLBs from human and CD-1 mice (P7) postnatal ganglionic gut were obtained and cultured as previously described by Torroglosa et al. [13]. Primary NLBs without subcultures were used, since secondary and tertiary NLBs acquire different properties compared with primary NLBs [35]. All the procedures involving mice were performed in accordance with the European Union guidelines (2010/63/EU) and the Spanish law (R.D. 53/2013 BOE 34/11370-420, 2013) related to the care and use of laboratory animals, and they were approved by the Animal Experimentation Ethics Committee (EAEC/IEC) of University Hospital Virgen del Rocío/Institute of Biomedicine of Seville (IBIS) (Project identification code: 1509-N-16 (December 2015), 2149-N-19 (December 2019) and 20191220134633-1 (October 2019), which complies with the tenets of the declaration of Helsinki). Human NLBs were extracted from 17 non-related patients diagnosed with isolated HSCR (L-HSCR = 1; S-HSCR = 16, male: female = 14:3). A total of 6 patients with other gastrointestinal disorders such as anorectal malformations and enterocolitis were considered as controls (male:female = 3:3). The age range of the subjects studied was 3 months to 3 years. Written informed consent for surgery, clinical, and molecular genetic studies was obtained from the guardians of all children.

### 4.2. ChIP-seq Assay

ChIP-seq assay was performed in EPCs derived from mouse-NLBs (10^6^). ChIP-seq and sequencing data analysis were carried out as previously described in detail in Villalba–Benito et al. [14]. A 10% of samples volume/reaction was kept as controls (input) before immunoprecipitation. Mouse IgG (Diagenode) was used as negative control (IgG) for ChIP assay in parallel with Anti-PAX6 antibodies (Millipore, Burlington, MA, USA) at 3 µg. A total of 3 pools of libraries, containing 16 ChIP, 16 inputs, and 4 IgG samples were sequenced in the MiSeq System (Illumina, San Diego, CA, USA). Bowtie was used to align ChIP-seq reads to mouse genome mm9. Model-based Analysis of ChIP-Seq (MACS) was used for calling the genomic regions significantly enriched (peaks). Overlapping peaks from IgG libraries with respect to input were eliminated from the analysis in order to ensure the quality of the results, and to obtain the specific target regions of PAX6. Regarding sequencing data analysis, all the FASTQ files from the ChIPs within the same pool were merged before using the MACS software. Then, BETA minus (http://cistrome.org/ap/root), PAVIS (http://manticore. niehs.nih.gov/pavis2/), and PAPST (https://github.com/paulbible/papst) tools were used to identify the genes located nearest to the regions represented by those peaks. Finally, we selected the genes obtained after the analyses of at least 2 of the 3 software in at least 2 of the 3 pools (Figure 4).

### 4.3. qRT-PCR

The expression pattern was analyzed in NLB cultures through the TaqMan® Array Plate (Life Technologies, Carlsbad, CA, USA) in the Applied Biosystems 7900HT system (Life Technologies, Carlsbad, CA, USA). Total RNA was isolated using RNeasy Micro kit (Qiagen, Hilden, Germany). Synthesis of cDNA was performed using PrimeScript RT Master Mix (Takara, Kusatsu, Japan). The results were analyzed using the RQ Manager Software (Life Technologies, Carlsbad, CA, USA) based on the comparative cycle threshold (Ct) (ΔΔCt) method. *GAPDH* was used as an endogenous control gene. Following the software recommendations, we considered expression when Ct values were less than 35.

### 4.4. Grb2 Small Hairpin RNA-Expressing Lentiviral Vectors and Infection of NLB Cultures

Lentiviral vectors that constitutively expressed a small hairpin RNA (shRNA) directed against Grb2 mRNA were used to knock-down their expression in mouse NLBs. The ShRNA non-target control was used as the control (Sigma Aldrich, St. Louis, MO, USA). Cultures were divided into 3 groups: Uninfected group (control), shRNA non-target control group (LV-off-target), and infected group (LV-*Grb2*). Both LV and LV-off-target conditions were infected with a combination of two different shRNAs at a multiplicity of infection of 2. *Grb2* expression was evaluated through SYBR Green method (Bio-Rad, Hercules, CA, USA).

### 4.5. Grb2-Knockdown Study in NLB Cultures

In order to evaluate the *Grb2*-knockdown effect on the neurogenesis and cell growth, the expression of Nestin, βIII-Tubulin, and p75 markers, as well as both the size and number of NLBs were analyzed as previously described in Torroglosa et al. [36]. 

### 4.6. Sequence Variants Analysis 

Potential susceptibility variants for HSCR were searched in the gene sequences with expression in human NLBs using WES data from 56 HSCR patients [15]. Mutational screening of *GRB2* was performed in our series of 267 isolated HSCR patients by Sanger sequencing (187 S-HSCR, 27 L-HSCR, 19 TCA, and 55 with phenotype not available) using an ABI PrismH3730 Genetic Analyzer and the SeqScape v2.5 software (Life Technologies, Carlsbad, CA, USA). 

### 4.7. Databases and in Silico Tools

Several tools and databases were used in order to perform a comprehensive analysis of the results. The human orthologs of the PAX6 target genes selected in mouse were searched using MGI (http://www.informatics.jax.org). Gene interactions of our target genes as well as their functions were obtained through the IPA tool and GeneMANIA database (http://genemania.org/) [37]. For de novo discovery of motifs in a set of sequences, we used the web application MEME-ChIP (http://meme-suite.org/tools/meme-chip) [38]. 

### 4.8. Statistical Analyses for Gene Expression Studies

Data were presented as the mean ± SEM (standard error mean) of values obtained from at least three experiments. Comparisons between values obtained in control NLBs and HSCR-NLBs were analyzed using the Student’s *t*-test. Differences were considered significant when the *p*-value ≤ 0.05.

## Figures and Tables

**Figure 1 ijms-21-09061-f001:**
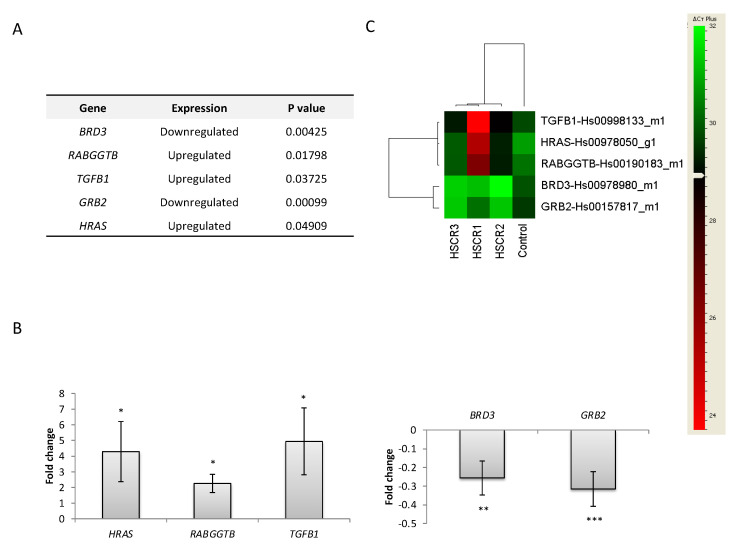
Genes with different expression levels in Hirschsprung disease (HSCR)-NLBs. (**A**) Genes that showed different expression levels between HSCR-NLBs and control NLBs. Analysis of differential gene expression of potential susceptibility genes for HSCR. (**B**) Analysis of differential gene expression relative to a fold change. (**C**) Heat map representative of the results. The heat map was generated using DataAssist v3.0 software (Life Technologies) and it represents the messenger RNA expression levels of such genes expressed in colon tissue from HSCR patients and controls. Genes were hierarchically clustered by Pearson correlation coefficient using average linkage. The color scale, representing ΔCt, is shown on the right side. Green indicates genes with relatively decreased expression in HSCR, whereas red indicates genes with relatively increased expression in HSCR compared with the controls. * *p* value < 0.05, ** *p* value < 0.01 and *** *p* value < 0.001.

**Figure 2 ijms-21-09061-f002:**
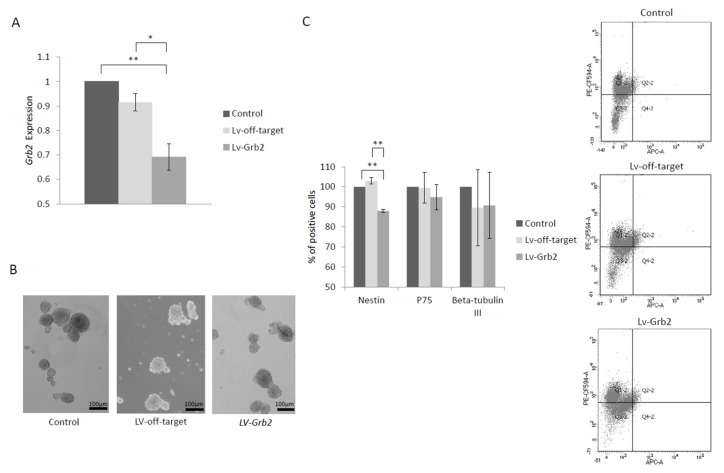
Functional role of *Grb2* in NLB cultures. (**A**) *Grb2* expression on mouse enteric precursor cells’ (EPCs) *Grb2* downregulation. Uninfected group (control), small hairpin RNA (shRNA) non-target control group (LV-off-target), and infected group (LV-*Grb2*). *p*-value control/LV-*Grb2* = 0.0017; LV-off-target/LV-*Grb2* = 0.02. (**B**) Effect of LV-*Grb2* on the size and number of NLB cultures. (**C**) Effects of LV-*Grb2* on the cell phenotypes derived from NLBs (EPCs that express Nestin, Beta-tubulin III, or p75 markers). For Nestin, *p*-value control/LV-*Grb2* = 0.002; LV-off-target/LV-*Grb2* = 0.003. A representative image of the decrease of Nestin-expressing cells (PE-CF594-A+) in LV-*Grb2* condition detected by flow cytometry is shown on the right side. Fluorochrome APC-A corresponds to Beta-tubulin III marker. * *p* value < 0.05, ** *p* value < 0.01 and *** *p* value < 0.001.

**Figure 3 ijms-21-09061-f003:**
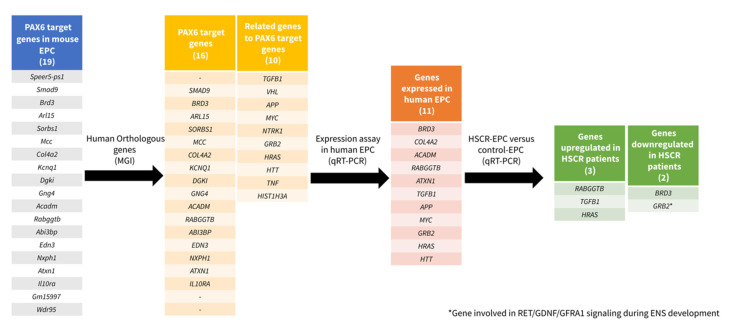
Diagram that shows the workflow with the selection, prioritization, and further validation of the candidate genes.

**Figure 4 ijms-21-09061-f004:**
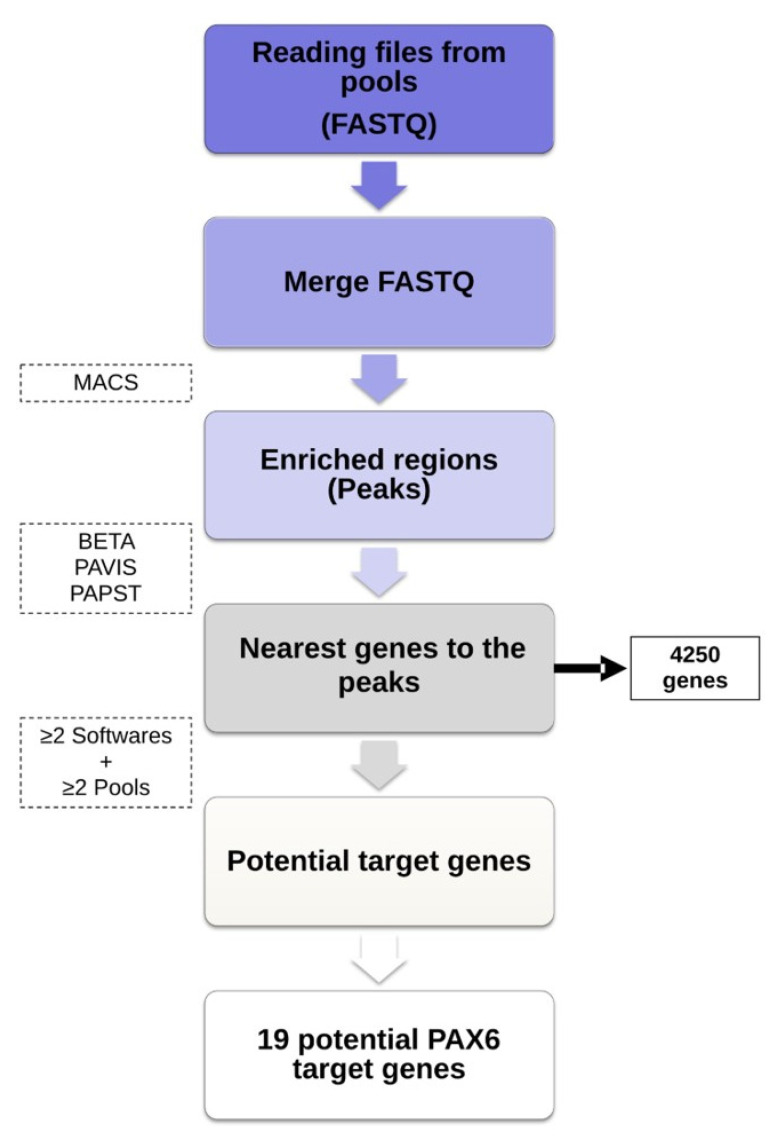
Diagram that shows the methodological approach used for the selection of PAX6 target genes.

**Table 1 ijms-21-09061-t001:** PAX6 Target Genes and Their Corresponding Peaks Within Genome Identified in Mouse Neurosphere-Like Bodies (NLBs) Through ChIP-Seq Assay.

Gene_Symbol	refseqID	Tss Distance	Chr	Start	End	Length	−10 * log10 (*p* value)	Fold_Enrichment	Experimental Replicate
**Speer5-ps1**	NR_001582	1774	chr10	43891861	43892191	330	80.89	31.34	1
		1714		43891774	43892159	385	84.05	54.84	2
***Smad9***	NM_019483	13586	chr3	54572813	54573365	552	54.2	19.01	1
		13553		54572951	54573162	211	177.55	42	2
**Brd3**	NM_001113573	1760	chr2	27332645	27333262	617	69.33	11.73	1
		1685		27332651	27333106	455	113.72	56.14	3
***Arl15***	NM_172595	−4637	chr13	114579840	114580317	477	66.57	24	2
		−4774		114579814	114580068	254	202.49	84.21	3
***Sorbs1***	NM_178362	−5037	chr19	40582865	40583513	648	329.19	118.82	2
		−5136		40582818	40583363	545	54.68	3.63	3
***Mcc***	NM_001085373	−32144	chr18	44939328	44940056	728	181.78	80.99	2
		−31615		44940046	44940397	351	120.18	50	3
***Col4a2***	NM_009932	39473	chr8	11352053	11352549	496	77.71	36.56	2
		39543		11352193	11352549	356	77.53	5.34	3
***Kcnq1***	NM_008434	78040	chr7	150371026	150371370	344	68.98	45.7	2
		78146		150371043	150371565	522	78.85	20	3
***Dgki***	NM_001081206	−92744	chr6	37156803	37157662	859	157.22	63.98	2
		−93040		37156803	37157070	267	106.97	15	3
***Gng4***	NM_001302997	−30831	chr13	13845260	13845728	468	70.97	18.11	1
		−30895		13845144	13845716	572	60.01	27.42	2
***Acadm***	NM_007382	−22702	chr3	153584775	153585035	260	99.69	45.7	2
		−22536		153584852	153585291	439	97.1	10	3
***Rabggtb***	NM_001163478	8975	chr3	153584775	153585035	260	99.69	45.7	2
		9141		153584852	153585291	439	97.1	10	3
***Abi3bp***	NM_001014422	−20897	chr16	56456644	56457478	834	96.15	30	2
		−21090		56456647	56457089	442	63.52	10	3
***Edn3***	NM_007903	−44835	chr2	174541128	174541718	590	72.63	45.7	2
		−44498		174541618	174541903	285	78.44	7.5	3
***Nxph1***	NM_008751	272432	chr6	9172169	9172731	562	57.91	26.12	1
		272487		9172167	9172843	676	95.92	54.84	2
***Atxn1***	NM_009124	123721	chr13	45936396	45936882	486	108.42	54.84	2
		123568		45936673	45936912	239	136.16	15	3
***Il10ra***	NM_008348	−6407	chr9	45070625	45071019	394	80.89	36.57	1
		−5195		45071866	45072203	337	68.59	45	2
***Gm15997***	NR_045423	−18653	chr5	150321687	150322218	531	60.47	13.46	1
		−18668		150321644	150322230	586	53.95	5	3
***Wdr95***	NM_029440	−9301	chr5	150321687	150322218	531	60.47	13.46	1
		−9316		150321644	150322230	586	53.95	5	3

**Table 2 ijms-21-09061-t002:** PAX6 DNA-Binding Motifs Recognized Within the PAX6 Binding Regions by MEME-ChIP Tool.

PAX6 Binding Motifs	Sites	Genes	E-value
CAYAYACACAYACACAHWCACACACACAYA	9	*Speer5-ps1, Atxn1, Sorbs1, Col4a2, Arl15, Abi3bp, Brd3, Wdr95/Gm15997, Edn3*	2.6 × 10^−50^
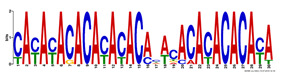			
GTATRTGTGTGTGT	11	*Col4a2, Arl15, Nxph1, Atxn1, Wdr95/Gm15997, Sorbs1, Acadm/Rabggtb, Abi3bp, End3, Gng4, Il10ra*	2.7 × 10^−12^
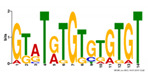			

**Table 3 ijms-21-09061-t003:** Human Selected Genes to Assess Their Expression Study in Human NLBs.

Selection Method of Genes
PAX6 Target Genes(ChIP-Seq)	Related Genes With PAX6 Target Genes (IPA)
Gene	Name	Gene	Name
*SMAD9*	*SMAD Family Member 9*	*TGFB1*	*Transforming Growth Factor Beta 1*
*BRD3*	*Bromodomain Containing 3*	*VHL*	*Von Hippel-Lindau Tumor Suppressor*
*ARL15*	*ADP Ribosylation Factor Like GTPase 15*	*APP*	*Amyloid Beta Precursor Protein*
*SORBS1*	*Sorbin And SH3 Domain Containing 1*	*MYC*	*MYC Proto-Oncogene, BHLH Transcription Factor*
*MCC*	*MCC Regulator of WNT Signaling Pathway*	*NTRK1*	*Neurotrophic Receptor Tyrosine Kinase 1*
*COL4A2*	*Collagen Type IV Alpha 2 Chain*	*GRB2*	*Neurotrophic Receptor Tyrosine Kinase 1*
*KCNQ1*	*Potassium Voltage-Gated Channel Subfamily Q Member 1*	*HRAS*	*Neurotrophic Receptor Tyrosine Kinase 1*
*DGKI*	*Diacylglycerol Kinase Iota*	*HTT*	*Huntingtin*
*GNG4*	*G Protein Subunit Gamma 4*	*TNF*	*Tumor Necrosis Factor*
*ACADM*	*Acyl-CoA Dehydrogenase Medium Chain*	*HIST1H3A*	*Histone Cluster 1 H3 Family*
*RABGGTB*	*Rab Geranylgeranyltransferase Subunit Beta*		
*ABI3BP*	*ABI Family Member 3 Binding Protein*		
*EDN3*	*Endothelin 3*		
*NXPH1*	*Neurexophilin 1*		
*ATXN1*	*Ataxin 1*		
*IL10RA*	*Interleukin 10 Receptor Subunit Alpha*

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
