# Peer review of "ChIP-Seq-Based Approach in Mouse Enteric Precursor Cells Reveals New Potential Genes with a Role in Enteric Nervous System Development and Hirschsprung Disease"

_ijms, 2020, doi:10.3390/ijms21239061_

Round 1
Reviewer 1 Report
Dear Authors,
very nice paper and moreover interesting.
I would confirm that this study findings represent a new contribution to advance in knowledge of the genetic basis of HSCR. Study like this are important to understand the molecular basis of Hirschsprung disease.
Author Response
Dear Reviewer,
Thank you so much for the revision of our article entitled “ChIP-seq-based approach in mouse Enteric Precursor Cells reveals new potential genes with a role in Enteric Nervous system development and Hirschsprung disease”. We really appreciate your comments and the time you took to review the article. We are delighted that you find our study as an important contribution to the advancement of knowledge in the molecular basis of HSCR.
Reviewer 2 Report
Hirschsprung disease (HSCR) is a condition that may arise from defects in the development of tissues containing cells commonly derived from the embryonic neural crest cell lineage.
This strongly suggest that intestinal tissues other than neural crest cell lineage are cytologically normal. The lack of the neural communication by enteric nervous syste is the basic cause of HSCR.
The main method of the present study by the authors are based on the generation of enteric neurosphere-like bodies (NLBs). Authors should clarify that the NLB contains neural crest cell lineage. Authors can not analyze HSCR without neural crest cell lineage.
Author Response
Dear Reviewer,
Thank you so much for the review of our research article entitled “ChIP-seq-based approach in mouse Enteric Precursor Cells reveals new potential genes with a role in Enteric Nervous system development and Hirschsprung disease”. We really appreciate your comments and the time you took to review the article. We apologize for not clarifying that NLBs contain neural crest-derived cells. Thanks to your comment, we have clarified this point in the manuscript. Please find attached the updated manuscript.

Round 2
Reviewer 2 Report
The manuscript has been revised well. This paper is an important contribution and I recommend that it will be accepted for publication.